# Impacts of structural properties of myosin II filaments on force generation

**Shihang Ding[1†], Pei-En Chou[2†], Shinji Deguchi[1]\*, Taeyoon Kim[3,4,5]\***

[1]Graduate School of Engineering Science, Osaka University, Toyonaka, Japan;
[2]School of Mechanical Engineering, Purdue University, West Lafayette, United States;
[3]Weldon School of Biomedical Engineering, Purdue University, West Lafayette, Japan;
[4]EMBRIO Institute, Purdue University, West Lafayette, United States; [5]Faculty of Science and Technology, Keio University, Yokohama, Japan

## eLife Assessment

The authors present a **useful** agent-based model to study the tensile force generated by myosin mini-filaments in actin systems (bundles and networks); by numerically solving a mechanical model of myosin II filaments, the authors provide insights into how the geometry of the molecular components and their elastic responses determine the force production. This work is of interest to biophysicists (in particular theoreticians) investigating force generation of motor molecules from a biomechanical engineering and physics perspective. The authors **convincingly** show that cooperative effects between multiple myosin filaments can enhance the total force generated, but not the efficiency of force generation (force per myosin) if passive cross-linkers are present. This work would benefit from a more extensive discussion of the physiological relevance of the results in view of the existing experimental literature, and how the principles that govern the behavior could be different for different motor proteins.

**\*For correspondence:**
deguchi@me.es.osaka-u.ac.jp (SD);
kimty@purdue.edu (TK)

[†]These authors contributed equally to this work

**Competing interest:** The authors declare that no competing interests exist.

**Abstract** Cells need intracellular forces for their physiological functions, such as migration, cytokinesis, and morphogenesis. The actin cytoskeleton generates a large fraction of the forces via interactions between cytoskeletal components, such as actin filament (F-actin), myosin, and actin cross-linking proteins. Myosin II plays the most important role in cellular force generation. Myosin II molecules self-assemble into filaments with different structures depending on myosin II isoforms and other conditions such as pH and ionic concentration. It has remained elusive how force generation in actomyosin structures is affected by the architecture of myosin II filaments. In this study, we employed an agent-based model to investigate the effects of the structural properties of myosin II filaments on force generation in disorganized actomyosin structures. We demonstrated that the magnitude of forces and the efficiency of force generation can vary over a wide range depending on the number and spatial distribution of myosin II filaments. Further, we showed that the number of myosin heads and the length of a bare zone at the center of myosin II filaments without heads highly affect the force generation process in bundles and networks. Our study provides insights into understanding the roles of the structural properties of myosin II filaments in actomyosin contractility.

## Introduction

Cells require forces for a wide variety of physiological functions, including cytokinesis, migration, and morphogenesis (_Lim et al., 2006_). It is well-known that mechanical forces are produced mainly by molecular interactions between filamentous actin (F-actin) and myosin II motor proteins in the actin cytoskeleton (_Salbreux et al., 2012_). Myosin II motors walk toward the barbed end of F-actin

using chemical energy stored in adenosine triphosphate (ATP). Myosin II molecules consist of two heads with a long tail, and they self-assemble into filamentous structures (*Craig and Woodhead, 2006*). There are three isoforms of myosin II: non-muscle, smooth muscle, and skeletal muscle myosins (*Thoresen et al., 2013*). These myosin isoforms form different filamentous structures whose length ranges from ~0.3 µm to ~1.5 µm with ~56 to ~800 heads (*Niederman and Pollard, 1975*; *Trombitás and Tigyi-Sebes, 1984*; *Tyska et al., 1999*; *Skubiszak and Kowalczyk, 2002*; *Oshima et al., 2012*; *Dasbiswas et al., 2018*; *Huang et al., 2021*; *Saito et al., 2021*). Unlike smooth muscle myosin forming a side-polar filament, non-muscle and skeletal muscle myosins form a bipolar filament with two sets of myosin heads located at both ends of the filament and a bare zone at its center whose length is ~160 nm. Although many studies have reported that the center bare zone provides binding sites for multiple molecules, its contributions to the force generation remain elusive. The number of myosin II molecules in the myosin filament also varies depending on conditions, such as pH level and ionic concentration (*Josephs and Harrington, 1966*; *Kaminer and Bell, 1966*; *Reisler et al., 1980*; *Pollard, 1982*). Individual myosin II heads have a relatively low duty ratio, meaning that they spend only a small fraction of their lifetime in the bound state unlike processive (i.e., high duty ratio) motors, such as myosin V or kinesin (*Kee and Robinson, 2008*). Thus, heads in a single myosin II molecule cannot walk along F-actin over a long distance by themselves. Myosin II circumvents this issue by forming the filamentous structure with multiple heads. If a myosin II filament has a sufficient number of heads, there are always a few myosin heads bound to F-actin, so the myosin II filament can remain proximal to the F-actin.

Heads with opposite polarities in the myosin II filament pull F-actins in opposite directions by walking toward the barbed ends of those F-actins, developing both tensile and compressive forces in disorganized actomyosin structures which are different from sarcomere found in muscle cells (*Murrell et al., 2015*). Theoretical and computational studies demonstrated that F-actin is easily buckled by compressive forces, leaving net tensile forces in disorganized bundles or networks that can mediate diverse contractile behaviors (*Lenz et al., 2012*; *Li et al., 2017*; *Okamoto et al., 2020*). Various reconstituted actomyosin systems have been employed to illuminate how contraction and force generation emerge from interactions between F-actins, myosin II filaments, and actin cross-linking proteins (ACPs) (*Mizuno et al., 2007*; *Bendix et al., 2008*; *Koenderink et al., 2009*; *Soares e Silva et al., 2011*; *Murrell and Gardel, 2012*; *Linsmeier et al., 2016*). Although they provided valuable insights, it has been understood poorly how various structural properties of myosin II filaments affect contraction or force generation in disorganized actomyosin structures.

Several computational models have been used to study the actomyosin contractility (*Åström et al., 2009*; *Alvarado et al., 2013*; *Stam et al., 2015*; *Alvarado et al., 2017*; *Cortes et al., 2020*; *Lenz, 2020*; *Weirich et al., 2021*). However, there were several limitations in the models. For example, some of the models treated myosin II filaments as either points (*Grewe and Schwarz, 2020*; *Lenz, 2020*), rods only with two binding sites (*Freedman et al., 2017*; *Chandrasekaran et al., 2019*; *Cortes et al., 2020*), or force dipoles (*Ronceray et al., 2019*). Such drastically simplified structures significantly differ from the real structure of the myosin II filament. In this study, we used our well-established agent-based model with detailed descriptions of the structure of myosin bipolar filaments (*Jung et al., 2015*; *Kim, 2015*; *Mak et al., 2016*; *Bidone et al., 2017*; *Li et al., 2017*; *Yu et al., 2018*), to illuminate how the number, length, bare zone size, and spatial distribution of myosin bipolar filaments influence force generation in disorganized actomyosin bundles and networks.

## Results
### Model overview
The detailed explanations about our model and all parameters used in the model are explained in *Supplementary files 1* and *Supplementary file 2*. Our model consists of only three key cytoskeletal elements – F-actin, motor, and ACPs – among >100 proteins found in the actomyosin structures of cells (*Liu et al., 2022*). They are simplified via cylindrical segments (*Figure 1A*); F-actin is represented by serially connected segments with 140 nm in length. Each ACP comprises two arms with 23.5 nm in length connected to its center point. To mimic the structure of bipolar filaments, each motor has a backbone, consisting of serially linked segments, and two arms on each endpoint of the backbone segments that represent 8 myosin heads ($N_h = 8$). The reference length of backbone segments ($L_{MB}$)

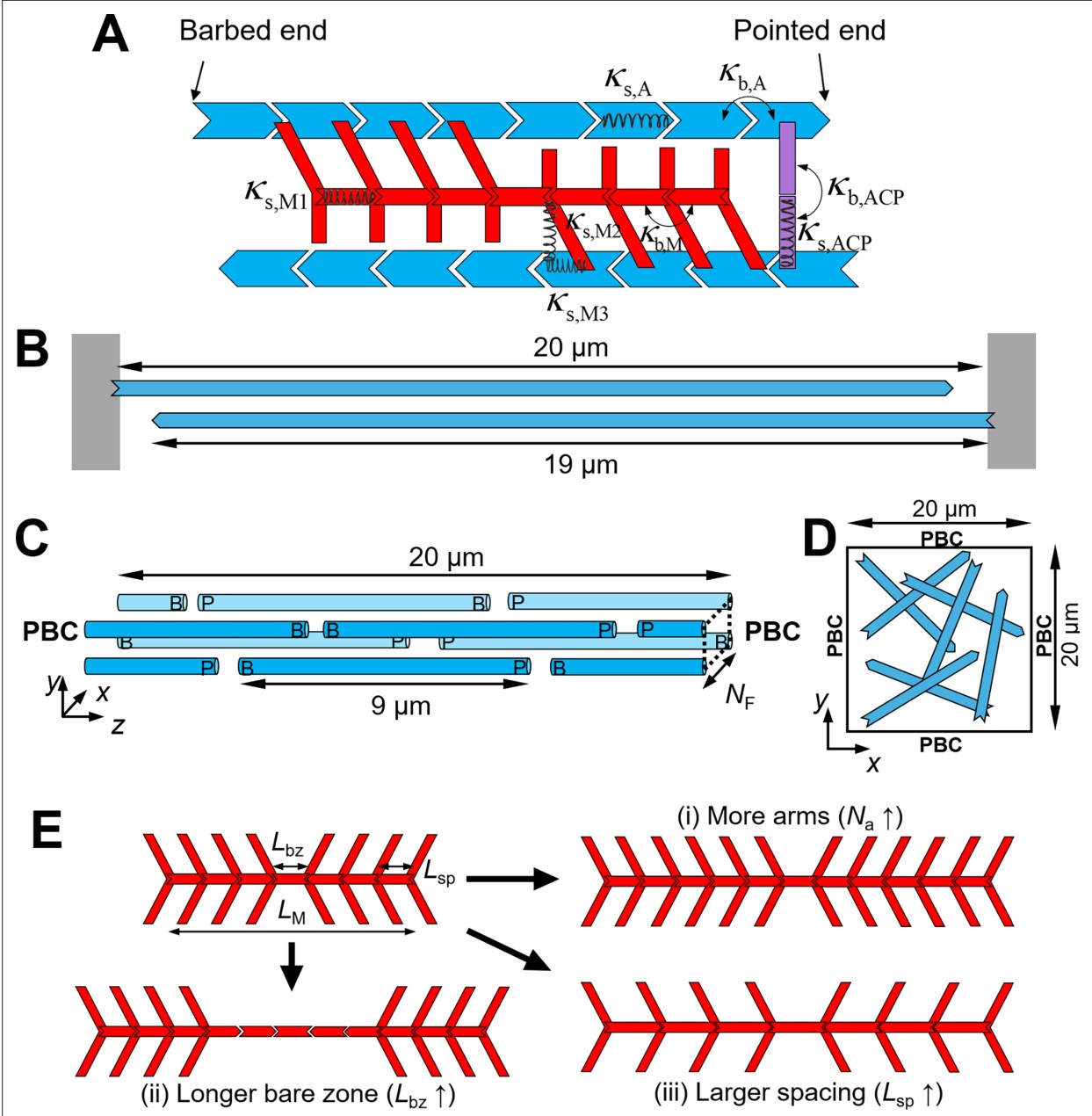

**Figure 1.** Modeling setups. (**A**) In the model, F-actin (blue), actin cross-linking protein (ACP, purple), and motor (red) are simplified by cylindrical segments. F-actin has polarity defined by barbed and pointed ends. Motors consist of a backbone with motor arms that can bind to and walk along F-actin. ACPs comprise two segments connected at the center point. $\kappa_s$ and $\kappa_b$ represent extensional and bending stiffnesses, respectively. (**B**) Two-filament system consisting of two F-actins whose barbed ends are clamped to rigid boundaries (gray). (**C**) The disorganized bundle system with $2N_F^2$ F-actins randomly located and oriented in the presence of the periodic boundary condition (PBC) in the z direction, where $N_F$ is a parameter defining bundle thickness. (**D**) The two-dimensional network system consisting of F-actins with random positions and orientations with the PBC in x and y directions. (**E**) A variation in the motor structure in three different ways: (i) increasing the number of motor arms ($N_a\uparrow$), (ii) increasing the bare zone length ($L_{bz}\uparrow$), and (iii) increasing a spacing between motor arms ($L_{sp}\uparrow$).

The online version of this article includes the following figure supplement(s) for figure 1:

**Figure supplement 1.** Measurement of the bundle force.

is 42 nm. The displacements of all the cylindrical segments at each time step are calculated by the Langevin equation and the forward Euler integration scheme. Deterministic forces in the Langevin equation include extensional and bending forces that maintain the equilibrium lengths of segments and equilibrium angles formed by segments, respectively, as well as a repulsive force exerted on overlapping pairs of actin segments for considering volume-exclusion effects.

Using the model, we simulate three types of systems: two filaments, bundles, and networks. In the two-filament simulations, a pair of anti-parallel F-actins with 19 µm in length are allocated in a rectangular computational domain (5 × 5 × 20 µm) with the periodic boundary condition (PBC) in $x$ and $y$ directions and the repulsive boundary condition in $z$ direction (**Figure 1B**). The barbed end of F-actins is clamped to two finite boundaries normal to the $z$ direction. In the bundle simulations, we employ the same rectangular domain with the PBC in all directions (**Figure 1C**). As in our previous study (**Kim, 2015**), we allocate F-actins in specific $x$ and $y$ coordinates. The number of possible positions for the allocation in $x$ or $y$ direction is defined by $N_F$, and spacing between adjacent coordinates in each direction is set to 27 nm. In each set of $x$ and $y$ positions, we position two F-actins with 9 µm in length in random $z$ position with random polarity. Thus, the total number of F-actins in the bundle is $2N_F^2$. In addition, the network simulations are conducted using a thin rectangular computational domain (20 × 20 × 0.1 µm) with the PBC in $x$ and $y$ directions and the repulsive boundary condition in $z$ direction (**Figure 1D**). In the domain, F-actins with ~10 µm in average length are allocated randomly in terms of positions and orientations as explained in supplementary text in detail. At the beginning of all simulations, while F-actins remain stationary, ACPs bind to F-actins to form permanent cross-linking points, and the arms of motors formed via the self-assembly of backbone segments bind to F-actins. After that, F-actins are allowed to move, and motor arms start walking and unbinding at force-dependent rates. Motor structures are varied in three different ways as explained later (**Figure 1E**).

## The distribution of motors and ACPs plays a key role in force generation

A previous in vitro study reported that tension developed in a thin bundle was almost directly proportional to the number of myosin heads in each motor, not to the number of motors (**Thoresen et al., 2013**). Although their experiments did not include any ACP, their theoretical explanation was based on an assumption that the bundle consists of serially connected contractile units like sarcomeres in muscle cells (**Thoresen et al., 2013**). They admitted that these contractile units do not have structural analogs in their disordered actomyosin bundles.

To verify their hypothesis, we first used a simple minimal model composed of 2 anti-parallel F-actins, 2 motors, and 16 ACPs (**Figures 1A and 2**). The number of arms per motor was set to $N_a$ = 24. We ran 20 simulations with random $z$ positions of motors and ACPs and found that the total force generated by the system, $F_{tot}$, fell into the following range (**Figure 2A**):

$$0.5F_M^{max} \leq F_{tot} \leq F_M^{max} \tag{1}$$

where $F_M^{max}$ is the maximal force that all motors can generate in this system:

$$F_M^{max} = \frac{1}{2}\sum_i^{N_M} F_{M,z}^i \tag{2}$$

where $F_{M,z}^i$ is the $z$ component of spring forces exerted by $i$th motor at a steady state, and $N_M$ is the number of motors. In this example, $N_M$ is 2. Note that only a quarter of motor arms (i.e., $N_a$ / 4) can bind to one F-actin due to two constraints assumed for the binding of the motor arms: motor arms can bind to F-actin when the arms are properly aligned with the polarity of F-actin, and two arms connected to the same point on a backbone cannot bind to the same F-actin. With two antiparallel F-actins, up to half of the motor arms can stay in the bound state. Thus, $F_{M,z}^i$ of a motor bound to two anti-parallel F-actins at the steady state is close to $F_{st}N_hN_a/2$, where $F_{st}$ is the stall force of one myosin head, and $N_h$ indicates the number of myosin heads represented by each motor arm. We found that a difference in $F_{tot}$ originated from the relative positions of motors and ACPs. When two motors were located in a very similar position with an almost full overlap by chance, $F_{tot}$ was close to $F_M^{max}$. When the two motors stayed apart without any overlap, $F_{tot}$ was close to either $0.5F_M^{max}$ or $F_M^{max}$ depending on whether or not there were ACPs between them. In the absence of ACPs between two

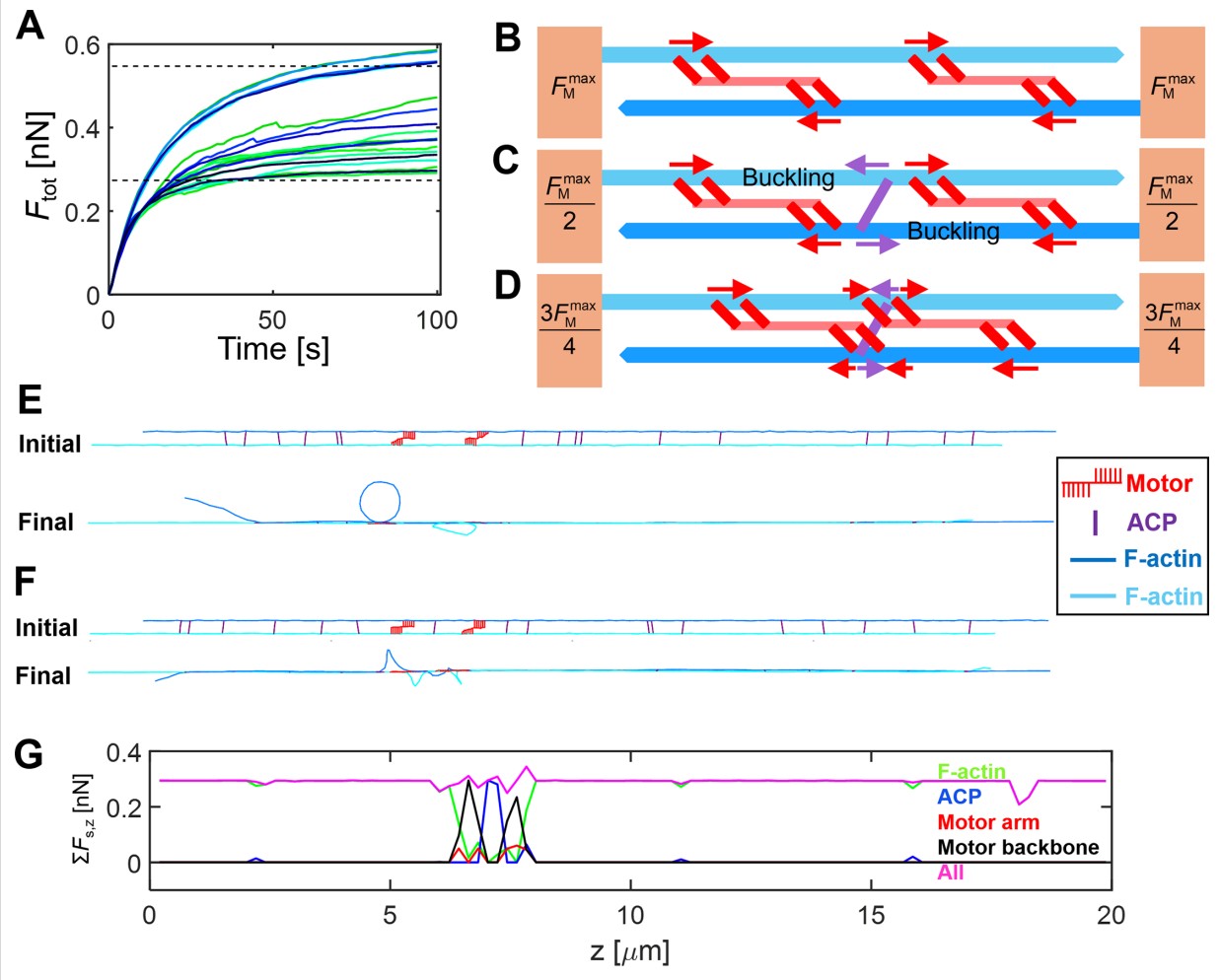

**Figure 2.** Interactions between motors and actin cross-linking proteins (ACPs) regulate force generation. (**A**) Time evolution of the force generated by two motors in the two-filament system. The upper and lower dashed lines indicate ideal upper and lower limits of a force that two motors can generate, respectively. Different colors represent distinct cases repeated 20 times. (**B**) Without any ACP between two motors, they can generate a force close to the upper limit which is twofold larger than a force that one motor can generate ($= F_M^{max}$). (**C**) With ACP(s) between two motors, ACPs counterbalance a force generated by one of the motors. Thus, they can generate a force close to the lower limit which is equal to the force that one motor can generate ($= F_M^{max}/2$). Part of F-actins is buckled due to two forces with opposite directions. (**D**) If two motors are close to each other, ACP can counterbalance a fraction of the force generated by one motor. Then, two motors can generate a force between the upper and lower limits. (**E, F**) Initial and final configurations (**E**) without or (**F**) with ACPs between two motors. The vertical dimension is increased 10 times to show the configurations clearly. (**G**) Measurement of tensile forces acting on F-actins (green), ACPs (blue), motor arms (red), motor backbones (black), or all (magenta) in $z$ direction.

motors, forces generated by the arms of two motors could add up to develop larger tensile forces on F-actins (*Figure 2B and E*). However, when ACPs existed between the two motors, it resulted in the buckling of F-actin between the ACPs and motor arms on one side, and counterbalanced the tension generated by motor arms on the other side (*Figure 2C and F*). As a result, F-actins ended up feeling tension corresponding to a force generated by one motor. These ACPs between two motors divide the bundle into serially connected contractile units as the assumption of the theoretical explanation in the other study mentioned earlier (*Craig and Megerman, 1977*; *Thoresen et al., 2013*). These ACPs were observed to experience larger tension than the rest of the ACPs since they directly counterbalanced large tension generated by motors (*Figure 2G*). There were quite a few cases showing an intermediate level of bundle tension between $0.5F_M^{max}$ and $F_M^{max}$ (*Figure 2A*). These cases had motors with a partial overlap and ACPs located within the $z$ range spanned by one of these motors. Due to the partial overlap with ACPs, forces generated by two motors partially added up, resulting in the intermediate tension (*Figure 2D*).

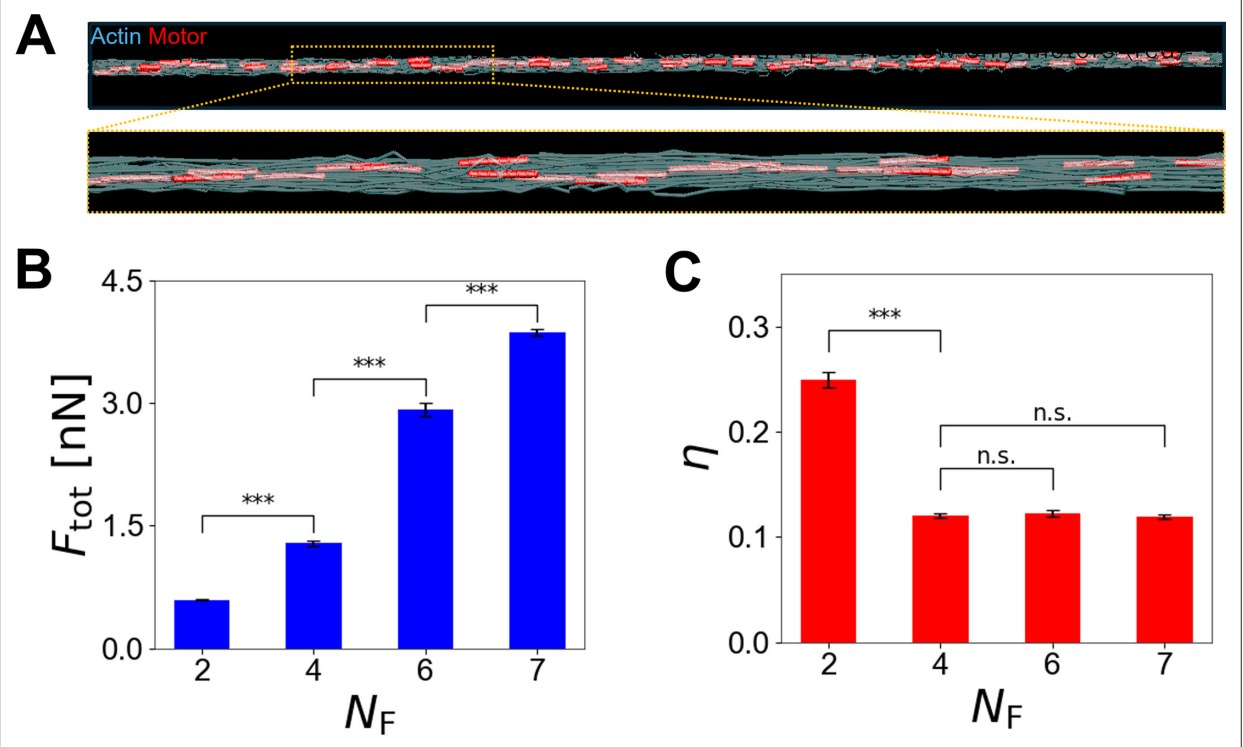

**Figure 3.** With fixed motor density, thicker bundles generate larger force in a less efficient manner. (**A**) An example of disorganized bundles with $N_F$ = 7 visualized at the beginning of the simulation, where $N_F$ is a parameter defining bundle thickness. F-actins are visualized as transparent elements to show the positions of motors. (**B**) Bundle-level force and (**C**) the efficiency of force generation measured at a steady state with different $N_F$. In thicker bundles ($N_F$ > 2), larger forces were generated, but the efficiency was lower than that in the thinnest bundle ($N_F$ = 2). Data represent the mean ± standard deviation (SD) calculated from n = 5 independent simulation runs. Statistical significance was assessed using paired t-test. Significance levels are denoted as follows: *** p ≤ 0.001, and n.s., not significant (p>0.05).

### Force generation in disorganized bundles is also regulated by the same mechanism

To verify the importance of the spatial distributions of motors and ACPs for force generation in more physiologically relevant structures, we repeated simulations using disorganized bundles with various sizes between $N_F$ = 2 (8 filaments) and $N_F$ = 7 (98 filaments) (*Figures 1B and 3A*). The densities of motors and ACPs were fixed at $R_{ACP}$ = 0.04 and $R_M$ = 0.005, resulting in $N_M$ ranging between 4 and 52. In all cases, each motor still had 24 arms ($N_a$ = 24). When the bundle became thicker by increasing $N_F$, a tensile force acting on the bundle ($F_{tot}$) also increased (*Figure 3B*). Because the motor density was fixed, more motors were present in thicker bundles, generating larger $F_{tot}$. In case of the thickest bundle ($N_F$ = 7), actin concentration was 12.25-fold higher (=98/8) compared to the thinnest bundle ($N_F$ = 2), so there were 13-fold more motors.

We defined the efficiency of force generation:

$$\eta = \frac{F_{tot}}{F_M^{max}} \tag{3}$$

In these bundles, $F_M^{max}$ is still defined by *Equation 2*, but $F_{M,z}^i$ was close to $F_{st}N_hN_a$ because there were more than one F-actin to bind for each polarity. Interestingly, $\eta$ was quite similar in cases with $N_F$ = 4, 6, and 7 (*Figure 3C*). By contrast, cases with $N_F$ = 2 exhibited much higher $\eta$ than the other cases. We probed why motors could generate forces in the thinnest bundle in a more efficient manner. With $N_F$ = 2, there were only four motors. The length of each motor with 24 arms is 462 nm (=42 nm × 11). The sum of the length of all four motors is only ~9% of bundle length, 20 μm, so they were less likely to significantly overlap with each other. Instead, most of them stayed apart with various distances. Given high ACP density, there were more than one ACP between adjacent motors. Thus, these motors

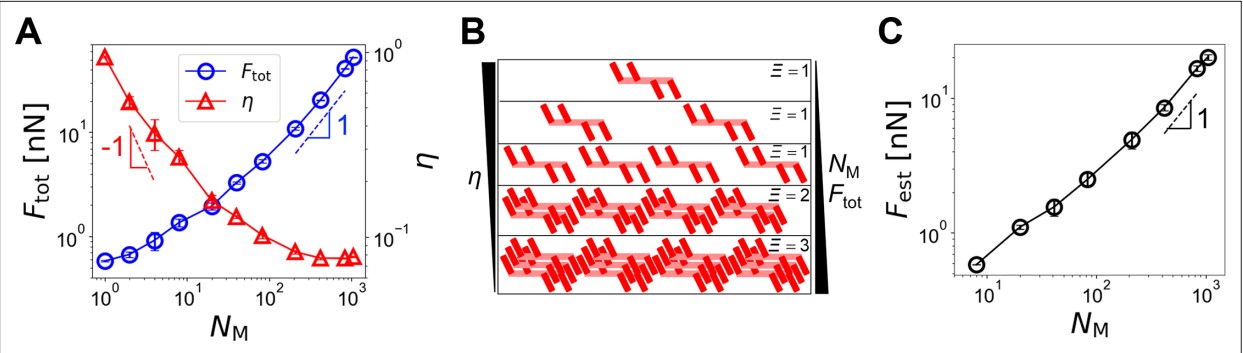

**Figure 4.** An increase in the number of motors ($N_M$) in disorganized bundles results in larger forces but smaller force generation efficiency. The thickest bundle ($N_F = 7$) was used for all cases. (**A**) Bundle-level force (blue circles) and the efficiency of force generation (red triangles) with a wide range of $N_M$ between 1 and 1045. With more motors, a larger force ($F_{tot}$) was generated, but the efficiency ($\eta$) was lower. (**B**) Configuration of motors with different $N_M$. $\Xi$ represents the estimated number of motors in the strongest contractile unit. With small $N_M$, $F_{tot}$ and $\Xi$ are unlikely to increase significantly until an entire bundle is occupied by motors, so $\eta$ is roughly 1 /$_{NM}$. By contrast, with high $N_M$, an increase $N_M$ directly enhances $F_{tot}$ and $\Xi$, and $\eta$ almost remains constant. (**C**) Prediction of a force using the positions of motors. To find the estimated force ($F_{est}$), $\Xi$ is calculated first, and **Equation 7** is used. Data represent the mean ± standard deviation (SD) calculated from n = 5 independent simulation runs.

The online version of this article includes the following figure supplement(s) for figure 4:

**Figure supplement 1.** Examples of calculating the maximal number of cooperatively overlapping motors.

formed separate contractile units, and their forces could not add up. Thus, $\eta$ for the thinnest bundle was roughly 1 /$_{NM}$ because $F_{tot}$ was close to the force generated by a single motor. Indeed, $\eta$ for the thinnest bundle was close to 0.25, confirming our rationale. If motors behave in the same manner in thicker bundles ($N_F > 2$), $\eta$ will be smaller than 0.25 because $N_M$ is proportional to $N_F^2$ due to fixed motor density. Although $\eta$ was actually smaller than 0.25 in thicker bundles, it was ~0.12 in all cases with $N_F > 2$, which was larger than the prediction, 1 /$_{NM}$. As the bundle was thicker, there was a higher chance for motors to overlap with each other or stay closely due to higher $N_M$. Then, some of these proximal motors could add up their forces if there is no ACP between them as discussed earlier. If the bundle has any set of such 'cooperative' motors, $F_{tot}$ can become larger than a maximal force generated by a single motor since $F_{tot}$ is determined by the largest force generated from one of the contractile units. As $N_M$ increases due to high $N_F$, the number of the cooperative motors tends to increase, resulting in larger $F_{tot}$. Thus, as $N_F$ increases, both $F_{tot}$ and $F_M^{max}$ increase, so $\eta$ can be higher than 1 /$_{NM}$ and rather insensitive to a change in $N_F$ when $N_F$ is not small.

## Bundles with more localized motors generate larger forces

Based on prior observations, overlaps between motors highly affect the force generation process. The extent of overlaps is determined by $N_M$ if the bundle length is fixed. To understand how force generation in the bundles depends on $N_M$ more systematically, we ran simulations with a wide range of $N_M$, using the thickest bundle ($N_F = 7$) and the same $R_{ACP} = 0.04$ and $N_a = 24$. The bundle tension, $F_{tot}$, showed a tendency to increase with higher $N_M$, but $\eta$ was inversely proportional to $N_M$ (**Figure 4A**). With low $N_M$, $F_{tot}$ was less sensitive to an increase in $N_M$, whereas $\eta$ was very sensitive to the increase in $N_M$. If there are a small number of motors in the bundle, adding a few more motors would not lead to a significant increase in $F_{tot}$ because the new motors are not likely to be located in positions where they can add up forces with other motors (**Figure 4B**). Instead, they will merely increase the number of contractile units, most of which have only one motor. Thus, as explained in **Equation 3**, $\eta$ was close to 1 /$_{NM}$. With higher $N_M$, $F_{tot}$ was sensitive to a change in $N_M$ because adding more motors to the bundle contributes to an increase in the number of motors in each contractile unit (**Figure 4B**). When $N_M$ was very high, $F_{tot}$ was almost directly proportional to $N_M$ with slope ~1 since all parts of the bundle were already occupied by motors, so adding more motors caused a direct impact on the magnitude of $F_{tot}$. In the range of high $N_M$, $\eta$ was much less sensitive to a change in $N_M$. For example, when $N_M$ was varied from 10 to 1000, $\eta$ was reduced to approximately half. The insensitivity of $\eta$ is attributed to an increase in both $F_{tot}$ and $F_M^{max}$ with higher $N_M$ as explained earlier. The plateau level of $\eta$ at ~0.08 is related to

the minimum number of motors required for saturating an entire bundle, implying that the plateau level would be higher if each motor is longer.

To verify this rationale, we developed a way to estimate the maximum number of overlapping motors, $\Xi$, using our simulation data:

$$\Xi = \max_{\substack{i=1\ldots N_M \\ j=L \text{ or } R}} \left( 1 + \sum_{k \neq i}^{N_M} \zeta_{ik}^{j} \right) \tag{4}$$

where $\xi_{ik}^{j}$ is

$$\xi_{ik}^{j} = \begin{cases} 1 & \text{if } L_c \leq L_{ov} \leq L_M \\ \dfrac{L_{ov}}{L_c} & \text{if } 0 < L_{ov} < L_c \\ 0 & \text{if } L_{ov} = 0 \end{cases} \tag{5}$$

$i$ and $k$ are the indices of motors, $j$ denotes the left or right side of a motor, and $L_{ov}$ is an overlap distance between two motors. In addition, $L_c$ is the critical length required for the cooperative overlap:

$$L_c = 2L_{sp} \left( \frac{N_a}{4} - 1 \right) \tag{6}$$

$L_c$ is twofold greater than the length occupied by motor arms on one side of a motor backbone. Two motors are considered a cooperative motor pair if $L_{ov}$ is equal to or greater than $L_c$ (*Figure 4—figure supplement 1*). With this cooperative overlap, ACPs cannot counterbalance forces exerted by motor arms. If $L_{ov}$ is shorter than $L_c$ but greater than zero, two motors are considered as a partially cooperative motor pair. By comparing all the motor combinations, $\Xi$ can be obtained. $\Xi$ can range between 1 (no overlap between motors) and $N_M$ (cooperative overlap between all motors). Using this $\Xi$, a force acting on the bundle can be estimated as $F_{est}$:

$$F_{est} = \frac{F_{st} N_h N_a \Xi}{2} \tag{7}$$

The sensitivity of $F_{est}$ to an increase of $N_M$ is high at large $N_M$ with a slope close to 1 (*Figure 4C*), which is consistent with our rationale described earlier. $F_{est}$ was generally smaller than $F_{tot}$ because this analysis does not account for actual bundle geometry consisting of multiple F-actins; if two motors are located far from each other in $x$ or $y$ direction, they may not counterbalance or add up forces. Nevertheless, we found that $F_{est}$ captures the overall dependence of $F_{tot}$ on parameters well.

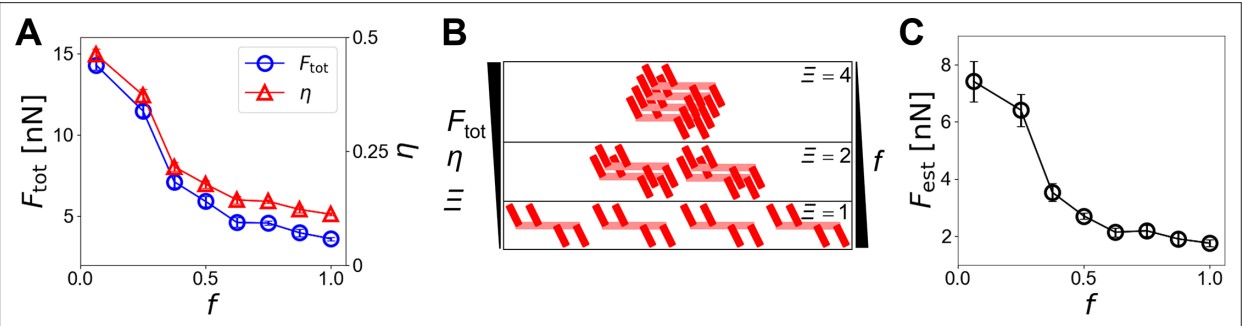

**Figure 5.** Motor distribution affects the force generation in disorganized bundles. We varied the relative size of a region where motors were initially located, $f$. $f=1$ means that motors can be located at any part of the bundle. (**A**) Bundle-level force ($F_{tot}$) and efficiency ($\eta$) depending on $f$. As motors were localized more closely to the center (i.e., smaller $f$), the force and the efficiency were higher. (**B**) Configuration of motors with different $f$. $\Xi$ indicates the estimated number of motors in the strongest contractile unit. Given the number of motors, smaller $f$ results in more cooperative overlaps (i.e., higher $\Xi$) and thus leads to higher $F_{tot}$ and $\eta$. (**C**) Prediction of the bundle-level force with different $f$. Data represent the mean ± standard deviation (SD) calulated from n = 5 independent simulation runs.

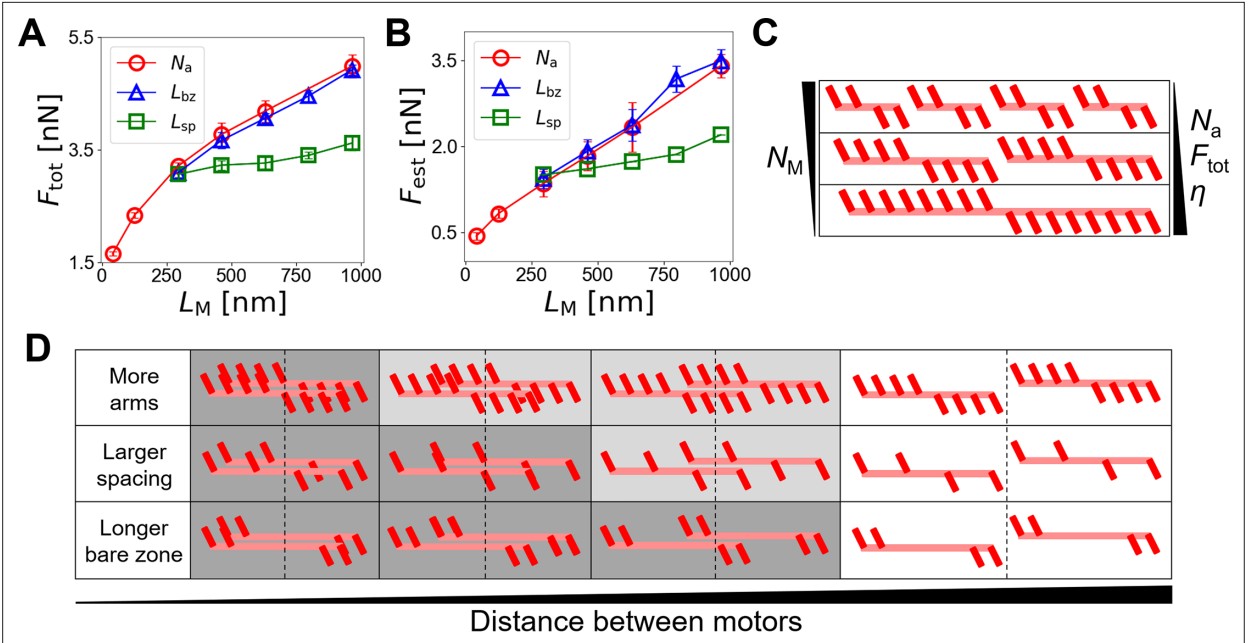

**Figure 6.** The architecture of motors impacts the force generation process in disorganized bundles. (**A**) Bundle-level force ($F_{tot}$) depending on $L_M$. The motor length ($L_M$) is varied between 42 nm and 966 nm by changing either of the number of motor arms ($N_a$, red circles) between 4 and 48, the bare zone length ($L_{bz}$, blue triangles) between 42 nm and 714 nm, or the spacing of motor arms ($L_{sp}$, green squares) between 42 nm and 138 nm. (**B**) Prediction of the bundle-level force using the positions of motors. (**C**) As each motor has more arms ($N_a\uparrow$), $F_{tot}$ and the efficiency of force generation ($\eta$) become higher because forces generated by motor arms are counterbalanced to a lesser extent. (**D**) Possible overlaps between two motors with different structures. A dark gray color indicates a fully cooperative overlap, and light gray indicates a partially cooperative overlap. To have the fully cooperative overlap, motors with many arms need to be located very closely, whereas motors with the long bare zone can overlap in the fully cooperative manner with a relatively long distance between them. Data represent the mean ± standard deviation (SD) calculated from n = 5 independent simulation runs.

The online version of this article includes the following figure supplement(s) for figure 6:

**Figure supplement 1.** Further analyses for cases with different motor length ($L_M$) varied by three methods.

So far, we have assumed that motors could be randomly located at any part of the bundle. If motors are confined within a smaller region, the extent of overlaps between motors (i.e., $\varXi$) could be increased with the same $N_M$, enhancing force generation. We tested the effects of motor distribution with $N_F = 7$, $R_{ACP} = 0.04$, $N_M = 52$, and $N_a = 24$. We allowed motors to be located only within a portion of the bundle defined by $f$ between 0 and 1. $f=1$ means that motors can be located anywhere. The center of the region for motor allocation is equal to the center of the bundle at $z=10$ μm. It was observed that a bundle generated the largest $F_{tot}$, and $\eta$ was the highest when motors were located near the center ($f=0.06$) (***Figure 5A***). Note that $F_M^{max}$ in the denominator of ***Equation 3*** is identical in these cases, meaning that $F_{tot}$ is directly proportional to $\eta$. Motors localized in relatively the same position are more likely to overlap in a cooperative manner (i.e., $L_{ov}\geq L_c$). Therefore, a large fraction of the motors were involved with the formation of a strong contractile unit (***Figure 5B***). As $f$ increased, motors were distributed on the bundle more sparsely. Then, many motors were partially overlapped or separated from each other. As a result, $F_{tot}$ and $\eta$ decreased (***Figure 5A***). The effect of a variation in $f$ on force generation was also reproduced well by the theoretical model explained earlier (***Figure 5C***).

## The structure of motors influences force generation in bundles

We have employed motors with 24 arms whose length is $L_M = 462$ nm with $L_{sp} = 42$ nm and $L_{bz} = 42$ nm. As mentioned earlier, the structure of myosin thick filaments can vary significantly, depending on myosin isoform and conditions. If motors are longer and have more arms, they may generate higher bundle forces. To verify this hypothesis, we tested cases with different $N_a$ between 4 and 48 and the thickest bundle ($N_F = 7$), with the total number of arms in the system, $N_a N_M$, fixed. With increasing $N_a$, $L_M$ was increased from 42 nm to 966 nm, but $N_M$ was decreased from 313 to 26 (***Figure 1E, i***). When $L_M$ was increased by higher $N_a$, $F_{tot}$ increased (***Figure 6A***, red). Because these cases had the same

total number of motor arms, $F_M^{max}$ in *Equation 3* was identical in all the cases. Thus, $\eta$ showed the same tendency as $F_{tot}$ (*Figure 6—figure supplement 1A*); with longer motors, $\eta$ was higher. The same tendency was observed in $F_{est}$ (*Figure 6B*, red). As motors have motor arms, individual contractile units become stronger, so $F_{tot}$ and $\eta$ are directly proportional to $L_M$ if there is no overlap between motors (*Figure 6C*). However, these motors are hard to overlap in a fully cooperative manner because $L_c$ is large (*Figure 6D*, top and *Equations 5; 6*). Thus, $\Xi$ was actually smaller with higher $L_M$ (*Figure 6—figure supplement 1B*, red). This explains why the dependence of $F_{tot}$ on $L_M$ was weaker at high $L_M$ than that at low $L_M$ (*Figure 6A*).

There are two additional ways to increase $L_M$ without a change in $N_a$ and $N_M$: increasing $L_{bz}$ at the center of motors or increasing $L_{sp}$ between motor arms uniformly (*Figure 1E, ii and iii*). We ran simulations with a variation in either $L_{sp}$ (between 42 nm and 138 nm) or $L_{bz}$ (between 42 nm and 714 nm), using the thickest bundle ($N_F = 7$) with $N_a = 16$. We compared results from these new simulations with those obtained with a variation in $N_a$ shown earlier. Interestingly, when $L_M$ was similar, $F_{tot}$ acquired with a change in $L_{bz}$ was similar to that with a variation in $N_a$ despite different $N_M$ (*Figure 6A*, blue), whereas $F_{tot}$ in cases with a change in $L_{sp}$ was noticeably lower (*Figure 6A*, green). $\eta$ showed an identical tendency as $F_{tot}$ because $F_M^{max}$ was the same in all the cases (*Figure 6—figure supplement 1A*).

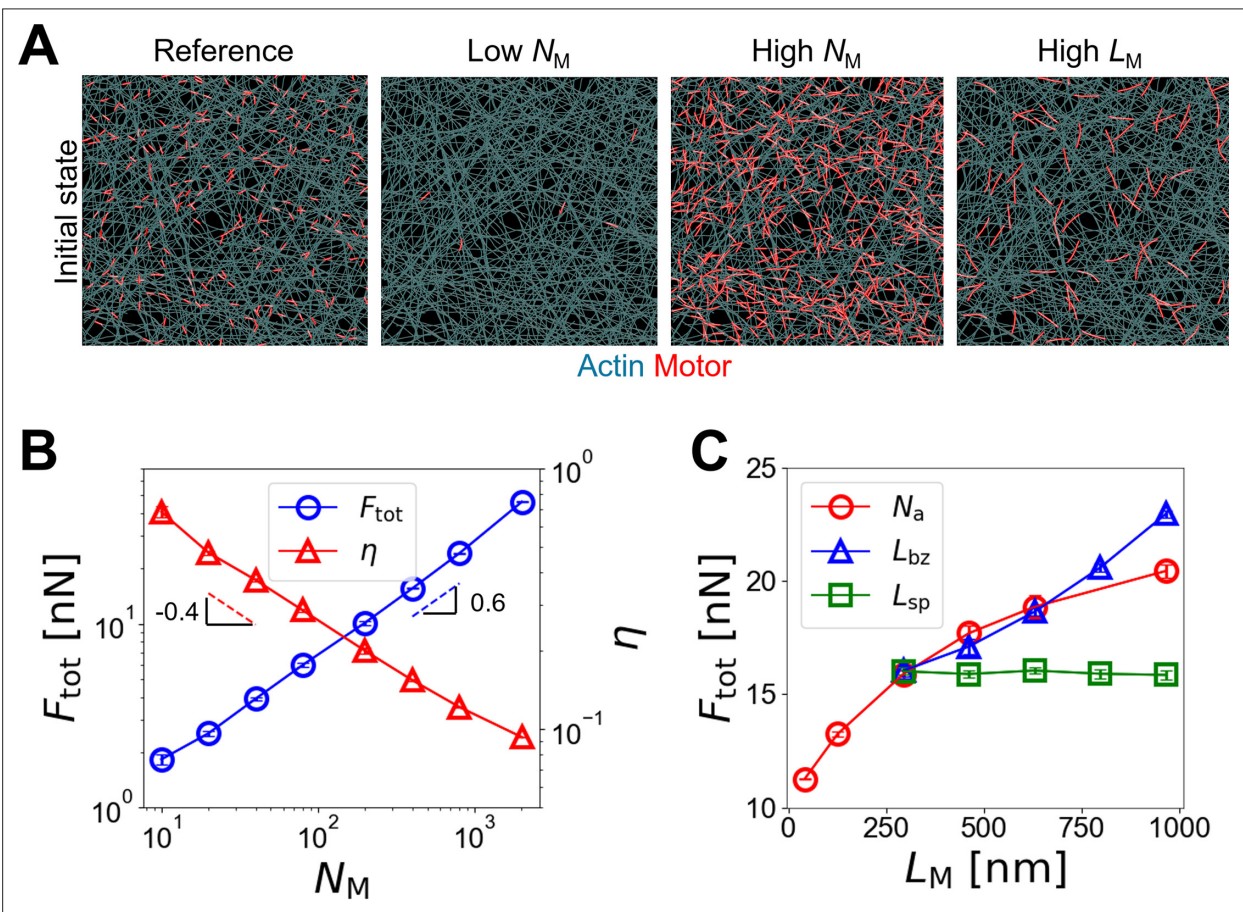

**Figure 7.** Force generation in two-dimensional actomyosin networks is governed by similar mechanisms. (**A**) Examples of networks at the initial state under the reference condition, with a smaller or larger number of motors, and with longer motors. These snapshots show only a quarter of networks to better visualize individual motors. (**B**) Network-level tension ($F_{tot}$) and the efficiency of force generation ($\eta$) with a different number of motors between 10 and 2006. (**C**) $F_{tot}$ depending on motor length varied by changing either of the number of motor arms ($N_a$, red circles) between 4 and 48, the bare zone length ($L_{bz}$, blue triangles) between 42 nm and 714 nm, or the spacing of motor arms ($L_{sp}$, green squares) between 42 nm and 138 nm. Data represent the mean ± standard deviation (SD) calculated from n = 5 independent simulation runs.

The online version of this article includes the following figure supplement(s) for figure 7:

**Figure supplement 1.** The efficiency of force generation in two-dimensional actomyosin networks with different motor length varied by three methods. Data represent the mean ± standard deviation (SD) calculated from n = 5 independent simulation runs.

$F_{est}$ also reproduced a similar tendency (**Figure 6B**). If $L_M$ is fixed, a longer bare zone makes motors have their arms near the two ends of their backbone. Then, motors can have a higher possibility to add up their forces because $L_c$ is smaller (**Figure 6D**, bottom and **Equations 5 and 6**). Thus, $\Xi$ was higher (**Figure 6—figure supplement 1B**, blue), resulting in higher $F_{tot}$ and $\eta$ despite smaller $N_a$. By contrast, motors with uniform large spacing between arms need to have more overlap to add up their forces (**Figure 6D**, middle). Smaller $\Xi$ and $N_a$ led to lower $F_{tot}$ and $\eta$ (**Figure 6—figure supplement 1B**, green).

## Force generation in actin networks is regulated by the same mechanism

To check the generality of our findings, we performed simulations using a two-dimensional (2D) network where F-actins and motors are randomly oriented without any bias, which is different from those in bundles (**Figure 1C**). We varied either of $N_M$, $N_a$, $L_{bz}$, or $L_{sp}$ as done for bundles (**Figure 7A**). $F_{tot}$ was smaller than the values measured in bundles under the same conditions due to the random orientations of motors (**Figure 7B and C**). Interestingly, we found that $F_{tot}$ is proportional to $N_M^{0.65}$, which is close to $\sqrt{N_M}$. Note that $F_{tot}$ was directly proportional to $N_M$ at large $N_M$ in case of the bundles (**Figure 3A**). Considering that motors are uniformly distributed on a 2D network, this weaker dependence on $N_M$ is expected; the average number of motors in each direction which can experience the cooperative overlap would be $\sim \sqrt{N_M}$. Maximal $N_M$ tested with the network was ~2500, so the dependence of $F_{tot}$ on $N_M$ with the network is similar to that with $N_M < \sim 50$ with the bundle (**Figure 4A**). $\eta$ showed similar magnitudes to those measured in the bundles (**Figure 6B**, **Figure 6—figure supplement 1**, and **Figure 7—figure supplement 1**). Note that $\eta$ was still calculated using **Equation 3**, but $F_M^{max}$ was obtained by summing the x or y component of forces exerted by all motors network and then averaging them:

$$F_M^{max} = \frac{1}{4}\left(\sum_i^{N_M} F_{M,x}^i + \sum_i^{N_M} F_{M,y}^i\right) \tag{8}$$

We found that the dependencies of $F_{tot}$ and $\eta$ on these parameters are similar to those observed with bundles, meaning that force generation regulated by cooperative overlaps between motors takes place even in networks. Although forces exerted by motors are not oriented in the same direction in networks, these forces can be counterbalanced or add up depending on the relative positions of motors.

## Discussion

Actomyosin contractility is well-conserved machinery for generating mechanical forces in animal cells, facilitating cytokinesis, cell migration, and tissue morphogenesis (**Murrell et al., 2015**). Myosin II, which is the most important molecular motor for cellular force generation, exists as a highly organized form called thick filaments. Myosin thick filaments have been studied extensively during recent decades. However, despite various forms of thick filaments found in different types of cells, the effects of their structural properties on force generation have not been understood well.

There was an in vitro study that employed thin disorganized actomyosin bundles consisting of a few F-actins with three myosin isoforms (skeletal muscle, smooth muscle, and non-muscle myosins) in order to investigate the effects of the size of myosin II filaments (**Thoresen et al., 2013**). The study showed that a tensile force developed in bundles is directly proportional to the number of myosin heads in each myosin II filament if the total number of myosin heads is identical. They explained this direct proportionality via a theoretical bundle model consisting of serially connected contractile units in which a single myosin II filament generates a tensile force. This is partially consistent with observations in our study. However, such an explanation can be applied only to a bundle without any overlap between thick filaments, meaning that there are only a few thick filaments.

Although force generation by actomyosin contractility has been investigated in several theoretical and computational studies, most of the previous models used drastically simplified motors without consideration of thick filament structures (**Freedman et al., 2017**; **Chandrasekaran et al., 2019**; **Eliaz et al., 2020**). Therefore, it was not feasible to investigate the importance of the structural properties

of thick filaments in force generation in those studies. There were a few models accounting for thick filament structures. However, only one thick filament was simulated (**Stam et al., 2015**; **Weirich et al., 2021**) due to high computational cost, or force generation process was not probed (**Cortes et al., 2020**).

In our study, using motors with the geometry of bipolar thick filaments, we systematically showed how the force generation process in disorganized actomyosin structures, including bundles and networks, is regulated by the number, spatial distribution, and structural properties of motors. First, using the simplest system with only two F-actins, we showed that motors with ACPs between them cannot add up their forces to generate a larger force (**Figure 2**). Then, we probed the force generation process in disorganized bundles with different thickness and found that the efficiency of force generation was lower in thicker bundles although the bundle-level force was higher (**Figure 3**). This was attributed to an increase in the number of motors in thicker bundles. We then found that a larger force was developed in the same bundle in a less efficient way when there were a larger number of motors (**Figure 4**). When motors were sparsely distributed without an overlap between them, the efficiency of force generation was inversely proportional to the number of motors (i.e., $\eta \propto 1/N_M$). As the number of motors further increased, motors started cooperatively overlapping with each other, forming stronger contractile units. Thus, the efficiency became greater than $1/_{NM}$. We also tested the effects of motor distribution on force generation to verify the importance of cooperative overlaps (**Figure 5**). As motors were distributed in a more confined region, bundles generated larger forces because there could be more cooperative overlaps between more densely distributed motors. We also found that force generation was enhanced and more efficient when motors were longer with more arms because forces generated by the arms of one motor can simply add up (**Figure 6**). In addition, longer motors with a long bare zone and a few arms could generate large forces in bundles than longer motors with a small bare zone and large spacing between arms since the former can achieve the cooperative overlap by a much smaller overlap (**Figure 6**). Our results imply that the consideration of F-actin connectivity may not be enough to accurately predict force generation unlike predictions from a recent study (**Eliaz et al., 2020**).

Although this study focused mainly on parameters related to motor structures, we expect that other parameters would affect the force generation process. For example, as we showed before (**Jung et al., 2015**; **Kim, 2015**), a decrease in ACP density would reduce forces by deteriorating connectivity between filaments. With very low ACP density, some of the neighboring motors may not have ACPs between them, thus adding up their forces as shown in **Figure 2**. However, such low ACP density may not maintain the structure of bundles or cross-linked networks well. In addition, the force-dependent unbinding of ACPs could change the spatial distribution of ACPs during force generation. If they behave as a slip bond which unbinds more frequently with higher forces, ACPs may not stay between two motors for a long time due to high tension (**Mulla et al., 2022**). Then, forces generated by two motors may have a higher chance to add up. By contrast, if they behave as a catch bond which unbinds less frequently with larger forces, more ACPs will be recruited between two motors, reducing a chance to add up forces. The length of actin filaments is unlikely to affect the force generation process significantly unless filaments are very short. Additionally, as we showed before (**Yu et al., 2018**; **Jung et al., 2019**), actin turnover would reduce forces by competing with motor activities, change connectivity between filaments over time, and prevent motors from being stalled for a long time, all of which could affect force generation.

Our findings can be applied to understand the structures of stress fibers that are known to mediate physiological processes, such as cell protrusion, cytokinesis, and cell shape maintenance (**Deguchi et al., 2006**; **Kaunas and Deguchi, 2011**). In stress fibers, non-muscle myosin II plays a main role in generating contractile forces. There are two types of contractile stress fibers: transverse arcs and ventral stress fibers (**Lehtimäki et al., 2021**). Ventral stress fibers, which are assembled from pre-existing stress fiber precursors, have a sarcomere-like structure with cascaded (i.e., serially connected) contractile units divided by $\alpha$-actinin which is one type of ACPs. By contrast, transverse arcs are smaller without distinct repeated structures with much fewer ACPs (**Lehtimäki et al., 2021**). ACPs in ventral stress fibers can counterbalance forces generated by motors as shown in **Figure 2**. The structure with cascaded units and many ACPs prevents the bundle-generated force from increasing beyond a force generated by a single contractile unit. Therefore, it is expected that ventral stress fibers would generate smaller forces than transverse arcs if their thickness is similar. However, the

bundle-generated force would be proportional to the thickness of bundles as shown in *Figure 3*. Thus, ventral stress fibers, which are typically thicker, are known to generate larger tension than transverse arcs (*Lee et al., 2018*).

Although we focused on force generation, the contractile behaviors of actomyosin structures (i.e., a decrease in length) have also been of great interest. Our model can be used to study such contractile behaviors by deactivating the PBC and removing connection between one end of bundle/network and a domain boundary as done previously (*Li et al., 2017*). To achieve higher contractile speed with the same total number of myosin heads, the existence of multiple contractile units would be better as suggested in a previous work (*Thoresen et al., 2013*). This means that there is a trade-off between force generation and contractile speed. Previous studies also showed that the contractile speed of networks is proportional to motor density (*Murrell et al., 2015*; *Yu et al., 2018*; *Malik-Garbi et al., 2019*). We may be able to use our model to systematically investigate how the contractile speed is regulated by parameters that we tested in this study, including the number, distribution, length, and structure of motors.

In conclusion, we probed how the force generation process in disorganized bundles and networks is regulated by the properties of myosin thick filaments. We found that more motors enable bundles and networks to generate larger tensile forces, but the efficiency of force generation was lower. With the same number of motors, forces generated by bundles and networks can vary to a large extent, depending on (i) whether they are sparsely or densely distributed, (ii) how many myosin heads each thick filament has, and (iii) how long the bare zone at the center is. Our results can be partly verified using different myosin isoforms under different conditions as done in the previous study (*Thoresen et al., 2013*). In addition, synthetic myosin thick filaments (i.e., one made with DNA origami [*Fujita et al., 2019*] whose structure can be designed artificially could be used to verify our findings).

## Materials and methods
### Brownian dynamics via the Langevin equation

In our model, F-actin consists of serially connected cylindrical segments with barbed and pointed ends. ACPs comprise two segments connected by an elastic hinge. Each motor mimics the structure of myosin thick filaments. Each motor has a backbone structure with a certain number of arms ($N_a$), and each of the arms represents eight real myosin heads ($N_h = 8$). Thus, the total number of myosin heads represented by one motor is $N_h N_a$. The motor backbone comprises several segments with identical length and has a bare zone at its center consisting of one or more segments. Motor arms are connected to the endpoints of the backbone segments which are not part of the bare zone. Spacing between adjacent motor arms corresponds to the length of one backbone segment between them.

The motions of the segments constituting F-actin, motor, and ACP are regulated by the Langevin equation with inertia neglected:

$$\mathbf{F}_i - \zeta_i \frac{d\mathbf{r_i}}{dt} + \mathbf{F}_i^{\mathrm{T}} = 0 \tag{9}$$

where $\mathbf{r}_i$ is a position vector of the $i$th element, $\zeta_i$ is a drag coefficient, $t$ is time, $\mathbf{F}_i$ is a deterministic force, and $\mathbf{F}_i^{\mathrm{T}}$ is a stochastic force satisfying the fluctuation-dissipation theorem (*Underhill and Doyle, 2004*):

$$\left\langle \mathbf{F}_i^{\mathrm{T}}\left(t\right) \mathbf{F}_j^{\mathrm{T}}\left(t\right) \right\rangle = \frac{2k_{\mathrm{B}}T\zeta_i\delta_{ij}}{\Delta t}\boldsymbol{\delta} \tag{10}$$

where $\delta_{ij}$ is the Kronecker delta, $\boldsymbol{\delta}$ is a second-order tensor, and $\Delta t = 1.15 \times 10^{-5}$ s is time step. Drag coefficients are defined via an approximated form for a cylindrical object (*Clift et al., 2005*):

$$\zeta_i = 3\pi\mu r_{\mathrm{c},i} \frac{3 + 2r_{0,i}/r_{\mathrm{c},i}}{5} \tag{11}$$

where $\mu$ is the viscosity of a surrounding medium, and $r_{0,i}$ and $r_{\mathrm{c},i}$ are the length and diameter of a segment, respectively. Position vectors of all the segments are updated at each time step via the Euler integration scheme:

$$\mathbf{r}_i \left( t + \Delta t \right) = \mathbf{r}_i \left( t \right) + \frac{\mathrm{d}\mathbf{r}_i}{\mathrm{d}t} \Delta t = \mathbf{r}_i \left( t \right) + \frac{1}{\zeta_i} \left( \mathbf{F}_i + \mathbf{F}_i^{\mathrm{T}} \right) \Delta t \tag{12}$$

## Structures of elements and deterministic forces

Deterministic forces account for (i) extensional forces that maintain equilibrium lengths, (ii) bending forces which maintain equilibrium angles, and (iii) repulsive forces representing volume-exclusion effects between F-actins. The bending and extensional forces originate from the following potentials:

$$U_{\mathrm{b}} = \frac{1}{2} \kappa_{\mathrm{b}} \left( \theta - \theta_0 \right)^2 \tag{13}$$

$$U_{\mathrm{s}} = \frac{1}{2} \kappa_{\mathrm{s}} \left( r - r_0 \right)^2 \tag{14}$$

where $\kappa_{\mathrm{b}}$ and $\kappa_{\mathrm{s}}$ are bending and extensional stiffnesses, $\theta$ and $\theta_0$ are instantaneous and equilibrium angles formed by adjacent segments, and $r$ and $r_0$ are the instantaneous and equilibrium lengths of cylindrical segments, respectively. An equilibrium angle formed by two adjacent actin segments ($\theta_{0,\mathrm{A}}$ = 0 rad) and the equilibrium length of actin segments ($r_{0,\mathrm{A}}$ = 140 nm) are regulated by the bending ($\kappa_{\mathrm{b,A}}$) and extensional ($\kappa_{\mathrm{s,A}}$) stiffnesses of F-actin, respectively. The value of $\kappa_{\mathrm{b,A}}$ corresponds to the persistence length of 9 μm (*Isambert et al., 1995*). An equilibrium angle formed by two arms of each ACP ($\theta_{0,\mathrm{ACP}}$ = 0 rad) and the equilibrium length of ACP arms ($r_{0,\mathrm{ACP}}$ = 23.5 nm) are maintained by the extensional ($\kappa_{\mathrm{s,ACP}}$) and bending ($\kappa_{\mathrm{b,ACP}}$) stiffnesses of ACPs, respectively.

An equilibrium angle formed by adjacent backbone segments ($\theta_{0,\mathrm{M}}$ = 0 rad) and the equilibrium length of motor backbone segments ($r_{\mathrm{s,M1}}$) are maintained by bending ($\kappa_{\mathrm{b,M}}$) and extensional ($\kappa_{\mathrm{s,M1}}$) stiffnesses, respectively. The value of $\kappa_{\mathrm{s,M1}}$ is equal to that of $\kappa_{\mathrm{s,A}}$, whereas the value of $\kappa_{\mathrm{b,M}}$ is larger than that of $\kappa_{\mathrm{b,A}}$ so that the backbone does not bend significantly. The reference value of $r_{\mathrm{s,M1}}$ is 42 nm, but it is varied in some of the simulations to increase the spacing between motor arms as well as the total length of the motor backbone. The number of segments in the bare zone determines the length of the bare zone. Two motor arms are located on each endpoint of backbone segments which are not part of the bare zone. The extension of each motor arm is regulated by the two-spring model with transverse ($\kappa_{\mathrm{s,M2}}$) and longitudinal ($\kappa_{\mathrm{s,M3}}$) springs. The transverse spring regulates an equilibrium distance ($r_{0,\mathrm{M2}}$ = 13.5 nm) between the endpoint of a motor backbone and an actin segment where the arm of the motor is bound, whereas the longitudinal spring maintains a right angle between the motor arm and the actin segment ($r_{0,\mathrm{M3}}$ = 0 nm).

Repulsive forces originate from a harmonic potential (*Kim et al., 2009*):

$$\mathrm{U_r} = \begin{cases} \frac{1}{2} \kappa_{\mathrm{r,A}} \left( r_{12} - r_{\mathrm{c,A}} \right)^2 & \text{if } r_{12} < r_{\mathrm{c,A}} \\ 0 & \text{if } r_{12} \geq r_{\mathrm{c,A}} \end{cases} \tag{15}$$

where $\kappa_{\mathrm{r,A}}$ is the strength of repulsive force, and $r_{12}$ is the minimum distance between two neighboring actin segments.

Forces exerted on actin segments by bound motor arms and ACPs or by the repulsive forces are distributed onto two ends (barbed and pointed ends) of the actin segments as described in our previous work in detail (*Jung et al., 2015*).

## Network assembly

Unlike F-actin in bundle simulations, F-actin in network simulations is formed by stochastic processes as in our previous studies. The formation of F-actin is initiated from a nucleation event with a constant rate constant, $k_{\mathrm{n,A}}$, with the appearance of one cylindrical segment in a random position with a random orientation perpendicular to the $z$ direction. The polymerization of F-actin is simulated by adding cylindrical segments at the barbed end of existing filaments with a rate constant, $k_{\mathrm{p,A}}$. The ratio of $k_{\mathrm{n,A}}$ to $k_{\mathrm{p,A}}$ is adjusted to result in the average filament length of ~10 μm. The rest of the assembly process is identical to that described in the main text.

## Dynamic behaviors of ACPs

The arms of ACPs bind to binding sites located on actin segments every 7 nm without any preference of contact angle at a constant rate. In all simulations, ACPs are not allowed to unbind from F-actins after binding, forming permanent cross-links.

## Dynamic behaviors of motors

The motor arms bind to binding sites on actin segments at the rate of $40N_h$ s$^{-1}$, where $N_h$ is the number of myosin heads represented by each motor arm as explained above. Motor arms can bind to F-actin only when they are properly aligned with respect to the polarity of F-actin, considering that myosin heads do not interact with F-actin if the heads are inappropriately oriented (*Sheetz and Spudich, 1983*; *Trombitás and Tigyi-Sebes, 1984*). Note that only with the proper alignment, motor arms are able to walk toward the end of the backbone where they are connected, not toward the center of the backbone. In addition, it is assumed that two motor arms connected to the same point of the backbone are not allowed to bind to the same F-actin. After binding, motor arms walk toward the barbed end of F-actin.

The walking ($k_{w,M}$) and unbinding ($k_{u,M}$) rates of the motor arms are defined by the parallel cluster model (PCM) to mimic the mechanochemical cycle of non-muscle myosin II (*Erdmann and Schwarz, 2012*; *Erdmann et al., 2013*). The details of the implementation and benchmarking of the PCM in our agent-based model are extensively described in our previous study (*Jung et al., 2015*). Note that $k_{w,M}$ and $k_{u,M}$ are smaller with a higher applied load with an assumption that motor arms show catch-bond behaviors. The unloaded walking velocity and stall force of motors are set to ~140 nm/s and $F_{st}N_h$, respectively, where $F_{st}$ is the stall force of one myosin head, ~5.7 pN.

## Variations in motor structures

In our study, we vary three structural properties of motors: the number of arms per motor ($N_a$), the length of the bare zone ($L_{bz}$), and spacing between motor arms ($L_{sp}$). We change the size of motors in three different ways. In the first way, $N_a$ is increased by connecting more segments with arms for a backbone, whereas $L_{bz}$ and $L_{sp}$ are equal to $L_{MB}$ (*Figure 1E, i*). In the second way, $L_{bz}$ is increased by adding more segments without arms to the bare zone, whereas $N_a$ is unchanged, and $L_{sp}$ is equal to $L_{MB}$ (*Figure 1E, ii*). In the third way, $L_{sp}$ and $L_{bz}$ are increased by making $L_{MB}$ higher, whereas $N_a$ is unchanged (*Figure 1E, iii*).

## Measurement of contractile forces

In the two-filament simulations and the bundle simulations, we measure tensile forces generated by the systems as follows. First, all segments crossing the cross-sections of the domain located every 200 nm in the *z* direction, which can be for either of F-actin, motor arm, motor backbone, or ACP arm, are identified (*Figure 1—figure supplement 1*). Then, the *z* component of spring forces acting on those segments is summed. Using these sums, four curves can be drawn as a function of *z* to show which segment supports more or less tensile loads in each *z* position. Note that at equilibrium, the sum of four forces is very similar regardless of *z* position, meaning that a large fraction of forces developed by motors act as spring forces rather than bending forces. The average of the sums calculated on all cross-sections at a steady state is considered the total force acting on the system, $F_{tot}$. In the network simulations, tensile forces are measured in a similar manner, using 20 cross-sections of the domain located evenly in the *x* and *y* directions. *x* or *y* component of tensile forces acting on all segments crossing each cross-section is summed at a steady state. The average of sums calculated on 40 cross-sections is considered $F_{tot}$.

## Acknowledgements

We gratefully acknowledge the support from EMBRIO Institute, contract #2120200, a National Science Foundation (NSF) Biology Integration Institute and the support from JST SPRING (JPMJSP2138 to SDi), JSPS (Japan Society for the Promotion of Science) KAKENHI (21H03796 to SDe).

## Additional information

### Funding

| Funder | Grant reference number | Author |
|---|---|---|
| National Science Foundation | 2120200 | Taeyoon Kim |
| Japan Science and Technology Agency | JPMJSP2138 | Shihang Ding |
| Japan Society for the Promotion of Science | 21H03796 | Shinji Deguchi |

The funders had no role in study design, data collection and interpretation, or the decision to submit the work for publication.

### Author contributions

Shihang Ding, Pei-En Chou, Formal analysis, Investigation, Visualization, Writing – original draft; Shinji Deguchi, Supervision, Funding acquisition, Writing – review and editing; Taeyoon Kim, Conceptualization, Software, Supervision, Funding acquisition, Project administration, Writing – review and editing

### Author ORCIDs

Shihang Ding ⓘ https://orcid.org/0000-0002-0279-4700
Taeyoon Kim ⓘ https://orcid.org/0000-0002-5588-8532

Reviewer #1 (Public review): https://doi.org/10.7554/eLife.105236.3.sa1
Reviewer #2 (Public review): https://doi.org/10.7554/eLife.105236.3.sa2
Author response https://doi.org/10.7554/eLife.105236.3.sa3

## Additional files

### Supplementary files

Supplementary file 1. List of parameters used in the model.

Supplementary file 2. List of parameter values used for adopting the 'parallel cluster model'. Note that we used slightly different values for $F_0$, $d$, and $k_m$ from those in the literature.

MDAR checklist

### Data availability

The current manuscript is a computational study, so no data have been generated for this manuscript. The source code is available on Github (copy archived at *Kim, 2025*).

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
