## [Editor Report · eLife Assessment]

The authors present a **useful** agent-based model to study the tensile force generated by myosin mini-filaments in actin systems (bundles and networks); by numerically solving a mechanical model of myosin II filaments, the authors provide insights into how the geometry of the molecular components and their elastic responses determine the force production. This work is of interest to biophysicists (in particular theoreticians) investigating force generation of motor molecules from a biomechanical engineering and physics perspective. The authors **convincingly** show that cooperative effects between multiple myosin filaments can enhance the total force generated, but not the efficiency of force generation (force per myosin) if passive cross-linkers are present. This work would benefit from a more extensive discussion of the physiological relevance of the results in view of the existing experimental literature, and how the principles that govern the behavior could be different for different motor proteins.

---

## [Referee Report · Reviewer #1 (Public review)]

Summary:

This work by Ding et al uses agent-based simulations to explore the role of the structure of molecular motor myosin filaments in force generation in cytoskeletal structures. The focus of the study is on disordered actin bundles which can occur in the cell cytoskeleton and can be investigated with in vitro purified protein experiments. A key finding is that the force generation depends on the number of myosin motor heads and the spatial distribution of the myosin thick filaments in relation to passive crosslinkers.

Strengths:

The work develops a model where the detailed structure of the myosin motor filaments with multiple heads is represented. This allows the authors to test the dependence of myosin-generated forces on the number of myosin heads and their spatial distribution.

The work highlights that forces from multiple myosin motors within a disordered actin bundle may not simply add up, but depend on their spatial distribution in relation to passive crosslinkers.

This may explain prior experimental observations in in vitro reconstituted actomyosin bundles that the tension developed in the bundle was proportional to the number of myosin motor heads per filament rather than the number of myosin filaments. More generally, this type of modeling can guide fundamental understanding of the relationship between structure and mechanical force production.

Weaknesses:

The work focuses on the structure of myosin filaments but ignores other processes that may determine contractility of actomyosin structures such as the dynamics of crosslinker binding/unbinding and actin polymerization/depolymerization.

The authors did not vary the relative concentration of myosin motors and passive crosslinkers. This would have revealed interesting competing effects between motor and crosslink density and distribution, that their model and other studies suggest are important.

Given the above factors and the lack of direct quantitative comparisons with the experiment, the physiological significance of the work remains hard to ascertain.

---

## [Referee Report · Reviewer #2 (Public review)]

Summary:

In this study, the authors use a mechanical model to investigate how the geometry and deformations of myosin II filaments influence their force generation. They introduce a force generation efficiency that is defined as the ratio of the total generated force and the maximal force that the motors can generate. By changing the architecture of the myosin II filaments, they study the force generation efficiency in different systems: two filaments, a disorganized bundle, and a 2D network. In the simple two-filament systems, they found that in the presence of actin cross-linking proteins motors cannot add up their force because of steric hindrances. In the disorganized bundle, the authors identified a critical overlap of motors for cooperative force generation. This overlap is also influenced by the arrangement of the motor on the filaments and influenced by the length of the bare zone between the motor heads.

Strengths:

The strength of the study is the identification of organizational principles in myosin II filaments that influence force generation. It provides a complementary mechanistic perspective on the operation of these motor filaments. The force generation efficiency and the cooperative overlap number are quantitative ways to characterize the force generation of molecular motors in clusters and between filaments. These quantities and their conceptual implications are most likely also applicable in other systems.

Weaknesses:

The detailed model that the authors present relies on over 20 numerical parameters that are listed in the supplement. Because of this vast number of parameters, it is not clear how general the findings are. On the other hand, it was not obvious how specific the model is to myosin II, meaning how well it can describe experimental findings or make measurable predictions. Although the authors partially addressed this point in the revisions, I still think it is not easy to see what are the fundamental principles that govern the behavior and how they could be different for different motor proteins.

The model seems to be quantitative, but the interpretation and connection to real experiments is rather qualitative in my point of view.

---

## [Author Response]

The following is the authors’ response to the original reviews

**Public Reviews:**

**Reviewer #1 (Public review):**
Summary:This work by Ding et al uses agent-based simulations to explore the role of the structure of molecular motor myosin filaments in force generation in cytoskeletal structures. The focus of the study is on disordered actin bundles which can occur in the cell cytoskeleton and have also been investigated with in vitro purified protein experiments.Strengths:The key finding is that cooperative effects between multiple myosin filaments can enhance both total force and the efficiency of force generation (force per myosin). These trends were possible to obtain only because the detailed structure of the motor filaments with multiple heads is represented in the model.

We appreciate your comments about the strength of our study.

Weaknesses:It is not clearly described what scientific/biological questions about cellular force production the work answers. There should be more discussion of how their simulation results compare with existing experiments or can be tested in future experiments.

Please see our response to the comment (1) below.

The model assumptions and scientific context need to be described better.

We apologize for the insufficient descriptions about the model and the scientific context. We revised the manuscript to better explain model assumptions and scientific context as described in our responses below.

The network contractility seems to be a mere appendix to the bundle contractility which is presented in much more detail.

Please see our response to the comment (6) below.

**Reviewer #1 (Recommendations for the authors):**
(1) It is not clearly described what scientific/biological questions about cellular force production the work answers. There should be more discussion of how their simulation results compare with existing experiments, or can be tested in future experiments. The authors do briefly mention Reference 4 where different myosin isoforms were used, but it is not clear that these experiments support the scalings predicted in this work in Figures 3-6. Also, the experiments in Ref. 4 apparently did not involve passive crosslinkers (ACPs) which are key in this study.

Thank you for the comment. In the 5th paragraph of the discussion section of the original manuscript, we applied our findings to understand how structural differences between ventral stress fibers and actin arcs could affect force generation. In addition, at the end of the discussion section, we mentioned that experiments with artificially-made myosin thick filaments could be used for verifying our results.

The experiments in Ref. 4 were only ones that we could directly compare our results with. In previous study, actomyosin bundles were experimentally created with ACPs (K.L. Weirich et al., Biophys J, 2021, 120: 1957-1970), but the motions of myosin thick filaments were only quantities measured in the experiments. In general, measuring forces generated by in vitro actomyosin bundles is very challenging. This is why the predictions from our model are particularly valuable for understanding the force generation of actomyosin structures.

(2) The architecture of the bundles seems to be prescribed by hand in these simulations. Several well-known stochastic aspects of the dynamics of actin and actin-binding proteins are not included in the model. For example, there is no remodeling of the actin structures through actin polymerization and depolymerization, or crosslink (ACP) binding and unbinding. Can the authors comment on why these effects could be neglected for the questions they want to address?

Thank you for the comment. We previously showed that the force generation process in actomyosin networks and bundles is affected by actin dynamics (Q. Yu et al., Biophys J, 2018, 115: 2003-2013) and the unbinding of ACPs (T. Kim, Biomech Model Mechanobiol, 2015, 14(2): 345-355 and W. Jung et al., Comput Part Mech, 2015, 2(4): 317-327).

However, we did not include the actin dynamics and the ACP unbinding in the current study to clearly understand the effects of the structural properties of thick filaments on the force generation process. We have learned that the stochastic behaviors of cytoskeletal components lead to noisier results, which requires us to run a much larger number of simulations to obtain statistically convincing data. We added the following paragraph in the discussion section of the revised manuscript:

“Although this study focused mainly on parameters related to motor structures, we expect that other parameters would affect the force generation process. For example, as we showed before, a decrease in ACP density would reduce forces by deteriorating connectivity between filaments. With very low ACP density, some of neighboring motors may not have ACPs between them, thus adding up their forces as shown in Figure 2. However, such low ACP density may not maintain the structure of bundles or cross-linked networks well. In addition, the force-dependent unbinding of ACPs could change the spatial distribution of ACPs during force generation. If they behave as a slip bond which unbinds more frequently with higher forces, ACPs may not stay between two motors for long time due to high tension. Then, forces generated by two motors may have a higher chance to add up. By contrast, if they behave as a catch bond which unbinds less frequently with larger forces, more ACPs will be recruited between two motors, reducing a chance to add up

forces. The length of actin filaments is unlikely to affect the force generation process significantly unless filaments are very short. Additionally, as we showed before, actin turnover would reduce forces by competing with motor activities, change connectivity between filaments over time, and prevent motors from being stalled for long time, all of which could affect force generation.”

(3) The present study is confined to the fixed density of motors and ACPs. However, these can be easily varied in in vitro experiments. Works such as Reference 4 show an optimum in contractility vs myosin concentration. Myosins act not only to slide actin filaments but also crosslink them.Can the authors vary myosin concentration to demonstrate such effects in their model?

As the reviewer pointed out, there is a belief that myosin thick filaments can serve as crosslinkers as well. However, unless there are a fraction of dead myosins (which remain bound on filaments without walking) or myosins dwell at the barbed ends filaments for very long time, it looks very hard for bundles or networks to generate large forces. A former experiment showed that active myosins increases the viscosity of actin networks, not elasticity (D. Humphrey et al., Nature, 2002, 416: 413-416) Computer simulations with reasonable assumptions did not show significant force generation without cross-linkers. We have tested systems with a large number of motors and a few cross-linkers in previous studies (T. Kim, Biomech Model Mechanobiol, 2015, 14(2): 345-355 and W. Jung et al., Comput Part Mech, 2015, 2(4): 317-327). We observed that large force/stress was generated momentarily, but it was relaxed very fast. It is expected that there will be similar outcomes if we try such conditions in the current study.

(4) Why is there a (factor of 1.5-2) discrepancy in the measured (Ftot) and estimated (Fest) force values in Figure 4-6? How can the authors improve their scaling arguments to capture this? What about the estimated efficiency?

Thank you for the comment. Indeed, there was a discrepancy between the actual and estimated forces. When the estimated force was calculated, we used the z positions of motors without consideration of the actual bundle geometry with multiple filaments. For example, if two motors are located on the opposite sides of the bundle (i.e., if they are located far from each other in x or y direction), forces generated by them may not counterbalance each other. Then, the estimated force can be smaller than the actual force because counterbalance between motors can be overcounted. The original manuscript had the following sentences to clarify this point: “*F*_est_ was generally smaller than *F*_tot_ because this analysis does not account for actual bundle geometry consisting of multiple F-actins; if two motors are located far from each other in *x* or *y* direction, they may not counterbalance or add up forces. Nevertheless, we found that *F*_est_ captures the overall dependence of *F*_tot_ on parameters well.”

(5) Several choices of parameter values used in the simulations are not clear:a) Why consider F actin of 140 nm specifically? Actin can come in a range of lengths. How do their results depend upon the length scale of actin?

It seems that there is a misunderstanding. 140 nm is the equilibrium length of one actin segment in our model. The actual F-actin consists of multiple actin segments. The length of Factin was 9 μm in bundle simulations and 10 μm (average) in network simulations. We expect that the general tendency of our results would not change with different filament length. However, if filament length becomes too short, the force generation process would be impaired due to lack of connectivity between filaments.

b) Similarly, very specific values of myosin backbone length (42 nm), number of myosin heads (8), number of arms (24), and Actin Cross-linking Proteins (ACPs). What informs these values and how will the results change if they are different? It is not especially clear how an "Arm" differs from "heads" and what kind of coarse-graining is involved.

In the “model overview” section of the original manuscript, we mentioned the following to clarify the definitions of motor arms and motor heads:

“To mimic the structure of bipolar filaments, each motor has a backbone, consisting of serially linked segments, and two arms on each endpoint of the backbone segments that represent 8 myosin heads (*N*_h_ = 8).”

We devised this coarse-graining scheme of myosin thick filaments in our previous work (T. Kim, Biomech Model Mechanobiol, 2015, 14(5): 1143-1155). Through extensive tests, we showed that force generation and motor behaviors are largely independent of coarse-graining level. In other words, a motor with the same value of *N*_h_*N*_a_ leads to similar outcomes regardless of the value of *N*_a_. However, in a bundle with multiple filaments, each motor has a sufficient number of arms to ensure simultaneous interactions with those filaments. This is why we decided to use*N*_h_ = 8 and *N*_a_ = 24.

To match the length of thick filaments and the total number of heads (*N*_h_*N*_a_) in the model with real myosin thick filaments, we have used 42 nm for each backbone length. Varying this length is equivalent to a variation in *L*_sp_ that we did for Figure 6.

We used high ACP density to ensure connections between all neighboring pairs of actin filaments. We already showed how the presence of ACPs affects the force generation process in Figure 2 using two actin filaments. It is expected that a variation of ACP density would affect our results to some extent. Since the main focus of the current study is the structural properties of motors, we did not explore the effects of ACP density. I hope that the reviewer would understand our intention.

(6) The manuscript focuses on disordered bundles with only one figure on networks. However, actin fibers also ubiquitously exist as disordered networks, and it is important to explore in more detail the contractile forces in such network arrangements.

We appreciate the comment. Because we plan to delve into the effects of motor structures on the force generation in networks as a follow-up study, we showed the minimal results in the current study to prove the generality of our findings. I hope that the reviewer would understand our intention and plan.

It is not described very clearly how these networks were generated.

We apologize for lack of explanation about how the networks were generated. We added the following section in Supplementary Text of the revised manuscript:

“Network assembly

Unlike F-actin in bundle simulations, F-actin in network simulations is formed by stochastic processes as in our previous studies. The formation of F-actin is initiated from a nucleation event with a constant rate constant, *k*_n,A_, with the appearance of one cylindrical segment in a random position with a random orientation perpendicular to the z direction. The polymerization of F-actin is simulated by adding cylindrical segments at the barbed end of existing filaments with a rate constant, *k*_p,A_. The ratio of *k*_n,A_to *k*_p,A_ is adjusted to result in the average filament length of ~10 μm. The rest of the assembly process is identical to that described in the main text.”

Crosslinked biopolymers like actin typically form disordered elastic networks with their coordination number below rigidity percolation threshold (z=4 in 2D), see for example review by Broedersz and Mackintosh Rev. Mod, Phys. 2013. Such networks should exist in the bendingdominated regime, where bending forces play a vital role in force propagation. Was that observed in the simulations? Why or why not?

We appreciate the comment. We are aware of the bending-dominated regime and indeed showed the importance of the bending stiffness of actin filaments at low shear strain level in our previous work (T. Kim et al., PLOS Comput Biol, 2009, 5(7): e1000439). In case of active networks with motors, such a bending-dominated regime has not been observed without external shear strain. Instead, buckling of actin filaments was found to be essential for breaking symmetry between tensile and compressive forces developed by motor activities. We have shown that the free contraction of networks is inhibited if filament bending stiffness is increased substantially (J. Li et al., Soft Matter, 2017, 13: 3213-3220 and T. Bidone et al., PLOS Comput Biol, 2017, 13(1): e1005277). We expect that contractile forces generated by bundles or networks will be reduced significantly if we highly increase bending stiffness. However, considering the focus of the current study is on the structural properties of motors, we did not perform such simulations.

(7) It would be interesting to see the simulated predictions of the bundle or network contraction dynamics. This can be done by changing to free boundary conditions so that the bundle can contract.

Thank you for the suggestion. We have previously investigated the free contraction of actomyosin networks with different motor density and ACP density (J Li et al., Soft Matter, 2017, 13: 3213). We observed that the rate of network contraction was higher with more motors and ACPs. However, we did not test the effects of the structural properties of thick filaments in the previous study. We plan to investigate the effects in future studies because the focus of the current study is the force generation process. Please note that in the discussion section of the original manuscript, we mentioned the following:

“Although we focused on force generation, the contractile behaviors of actomyosin structures (i.e., a decrease in length) have also been of great interest. Our model can be used to study such contractile behaviors by deactivating the periodic boundary condition and removing connection between one end of bundle/network and a domain boundary as done previously [20]. To achieve higher contractile speed with the same total number of myosin heads, the existence of multiple contractile units would be better as suggested in a previous work [4]. This means that there is a trade-off between force generation and contractile speed. Previous studies also showed that the contractile speed of networks is proportional to motor density [18, 43, 51]. We may be able to use our model to systematically investigate how the contractile speed is regulated by parameters that we tested in this study, including the number, distribution, length, and structure of motors.”

Minor suggestions for improvement:(1) What are the vertical markers in Figures 1E and F? They should be labelled. if they are crosslinkers, it is not clear why the color is different from Figure 1A and B.

We believe that the reviewer meant Figures 2E, F. Those vertical lines are indeed ACPs (crosslinkers). We changed the color of ACPs in Figure 1A and Figures 2B-D to purple to be consistent. In addition, we changed the colors of two filaments in Figs. 2B-D slightly to be consistent with Figure 2E.

(2) To help understanding, please include a figure showing how forces are measured.

We added Figure 1 - figure supplement 1 in the revised manuscript to explain how the bundle force is calculated.

(3) It should be possible to extend the scaling arguments to predict what is the crossover myosin density (N_M) in Figure 4a at which the efficiency changes from going as 1/N_M to saturating.

As the reviewer might have observed, the slope of the efficiency in Figure 4A gradually changes, rather than showing a sharp transition. Thus, it is hard to define one crossover myosin density.

Similarly, what are the slopes in Figure 6a-b?

We drew the reference lines in those two plots. Unfortunately, we do not have explanations about the origin of these slopes.

(4) Some more explanation for the observed values should be added. Figure 4: Why does efficiency plateau at a value close to 0.8 in (A)?

We assume that the reviewer meant the plateau of *η* close to 0.08, not 0.8. Our speculation for the origin of this plateau value is related to *L*_M_ (=462 nm under the reference condition). Ideally, ~43 motors are required to cover the entire length of the bundle (=20 μm). Under this condition, *η* is ~0.023. Although this is not 0.08, we believe that these two values are related to each other. For example, if we increase *L*_M_, this plateau level would increase. We added the following sentences in the result section of the revised manuscript:

“The plateau level of *η* at ~0.08 is related to the minimum number of motors required for saturating an entire bundle, implying that the plateau level would be higher if each motor is longer.”

Figure 5: Overlapping between motors seems to increase the total force applied by them because of cooperative effects. However, it is not abundantly clear why that should peak at a value of f = 0.06.

As shown in Figure 5B, smaller *f* always results in higher *F*_tot_ due to higher level of cooperative overlap. The minimum value of *f* we tested in this study was 0.06, so *F*_tot_ was maximal at *f* = 0.06.

(5) Why is the network force expected to scale approximately as sqrt(N_M)? Is it because of the 2D geometry where the number of motors along the x or y-direction scale as sqrt(N_M)?

We initially thought that the weaker dependence of the total force on *N*_M_ was related to the random orientations of motors. However, if the network is fully saturated with motors, the inclusion of more motors will increase forces in both x and y directions almost linearly, resulting in the direct proportionality of *F*_tot_ to *N*_M_. Our new hypothesis for weaker dependence is consistent with the reviewer’s speculation; the network is not fully saturated even with 1000 motors, so the entire regime shown in Figure 7B corresponds to that with *N*_M_ < 100 in Figure 4A where similar weaker dependence on *N*_M_ was observed. We added the following sentence in the result section of the revised manuscript to clarify this point:

“the average number of motors in each direction which can experience the cooperative overlap would be ~\begin{document}$\sqrt{N_{\mathrm{M}}}$\end{document}. Maximal *N*_M_ tested with the network was ~2,500, so the dependence of *F*_tot_ on *N*_M_ with the network is similar to that with *N*_M_ < ~50 with the bundle (Figure 4A).”

(6) Figures 6 D and A: Figure 6D suggests that there is a more full overlap in the cases where there was a longer bare zone or larger spacing between motor arms. However, the quantification of the total force in A shows that the force is highest for the case where LM was increased by increasing the number of arms. Why do the authors think that is? I would expect from the explanation in Fig 6D that the Lsp and Lbz would be higher than Na in Fig 6A.

Figure 6D shows a difference in the level of the cooperative overlap (\begin{document}$\xi_{i k}$\end{document}) between two motors. As the reviewer pointed out, the case with more arms shows the lowest \begin{document}$\xi_{i k}$\end{document}, resulting in the lowest \begin{document}$\Xi$\end{document} as we showed in Figure 4 - figure supplement 1B. However, as show in in Equation 7, the total force is a function of both *L*_a_ and *N*_a_ and lower \begin{document}$\Xi$\end{document}. Thus, due to higher *N*_a_ can be similar to that in the case with different \begin{document}$\Xi$\end{document}, the force in the case with different *L*_bz_. In the original manuscript, we had the following sentence to explain how the force can be similar between the two cases:

“Thus, \begin{document}$\Xi$\end{document} was higher (Figure 4 - figure supplement 1B, blue), resulting in higher *F*_tot_ and *η* despite smaller *N*_a_.”

**Reviewer #2 (Public review):**
Summary:In this study, the authors use a mechanical model to investigate how the geometry and deformations of myosin II filaments influence their force generation. They introduce a force generation efficiency that is defined as the ratio of the total generated force and the maximal force that the motors can generate. By changing the architecture of the myosin II filaments, they study the force generation efficiency in different systems: two filaments, a disorganized bundle, and a 2D network. In the simple two-filament systems, they found that in the presence of actin crosslinking proteins motors cannot add up their force because of steric hindrances. In the disorganized bundle, the authors identified a critical overlap of motors for cooperative force generation. This overlap is also influenced by the arrangement of the motor on the filaments and influenced by the length of the bare zone between the motor heads.Strengths:The strength of the study is the identification of organizational principles in myosin II filaments that influence force generation. It provides a complementary mechanistic perspective on the operation of these motor filaments. The force generation efficiency and the cooperative overlap number are quantitative ways to characterize the force generation of molecular motors in clusters and between filaments. These quantities and their conceptual implications are most likely also applicable in other systems.

Thank you for the comments about the strength of our study.

Weaknesses:The detailed model that the authors present relies on over 20 numerical parameters that are listed in the supplement. Because of this vast amount of parameters, it is not clear how general the findings are. On the other hand, it was not obvious how specific the model is to myosin II, meaning how well it can describe experimental findings or make measurable predictions. The model seems to be quantitative, but the interpretation and connection to real experiments are rather qualitative in my point of view.

As the reviewer mentioned, all agent-based computational models for simulating the actin cytoskeleton are inevitably involved with such a large number of parameters. Some of the parameter values are not known well, so we have tuned our parameter values carefully by comparing our results with experimental observations in our previous studies since 2009.We were aware of the importance of rigorous representation of unbinding and walking rates of myosin motors, so we implemented the parallel cluster model, which can predict those rates with consideration of the mechanochemical rates of myosin II, into our model. Thus, we are convincing that our motors represent myosin II.

In our manuscript, our results were compared with prior observations in Ref. 4 (Thoresen et al., Biophys J, 2013) several times. In particular, larger force generation with more myosin heads per thick filament was consistent between the experiment and our simulations.

Our study can make various predictions. First, our study explains why non-muscle myosin II in stress fibers shows focal distributions rather than uniform distributions; if they stay closely, they can generate much larger forces in the stress fibers via the cooperative overlap. Our study also predicts a difference between bipolar structures (found in skeletal muscle myosins and nonmuscle myosins) and side polar structures (found in smooth muscle myosins) in terms of the likelihood of the cooperative overlap. As shown below, myosin filaments with the bipolar structure can add up their forces better than those with the side polar structure when their overlap level is the same.

It was often difficult for me to follow what parameters were changed and what parameters were set to what numerical values when inspecting the curve shown in the figures. The manuscript could be more specific by explicitly giving numbers. For example, in the caption for Figure 6, instead of saying "is varied by changing the number of motor arms, the bare zone length, the spacing between motor arms", the authors could be more specific and give the ranges: "is varied by changing the number of motor arms form ... to .., the bare zone length from .. to..., and the spacing between motor arms from .. to ..".This unspecificity is also reflected in the text: "We ran simulations with a variation in either *L*_sp_ or *L*_bz_" What is the range of this variation? "When*L*_M_ was similar" similar to what? "despite different *N*_M_." What are the different values for *N*_M_? These are only a few examples that show that the text could be way more specific and quantitative instead of qualitative descriptions.

We appreciate the comment. In the revised manuscript, we specified the range of the variation in each parameter.

In the text, after equation (2) the authors discuss assumptions about the binding of the motor to the actin filament. I think these model-related assumptions and explanations should be discussed not in the results section but rather in the "model overview" section.

Thank you for pointing this out. In the original manuscript, we described all the details of the model in Supplementary Material. We feel that the assumptions about interactions between motors and actin filaments are too detailed information to be included in the model overview section.

The lines with different colors in Figure 2A are not explained. What systems and parameters do they represent?

The different colors used in Figure 2A were used for distinguishing 20 cases. We added the explanation about the colors in the figure caption in the revised manuscript.

**Reviewer #2 (Recommendations for the authors):**
To guarantee the reproducibility of the results, I recommend that the authors publish their simulation code on GitHub.

We appreciate the reviewer’s suggestion. Following the suggestion, we prepared and posted the code on GitHub as mentioned in the Data Availability of the revised manuscript: The source code of our model is available on GitHub: https://github.com/ktyman2/ThickFilament”